

# V3Geo: A cloud-based repository for virtual 3D models in geoscience

Simon J. Buckley[1], John A. Howell[2], Nicole Naumann[1], Conor Lewis[1], Magda Chmielewska[2], Kari Ringdal[1], Joris Vanbiervliet[1], Bowei Tong[1], Oliver S. Mulelid-Tynes[3], Dylan Foster[3], Gail Maxwell[2], Jessica Pugsley[2]

[1]NORCE Norwegian Research Centre, P.O. Box 22 Nygårdstangen, N-5838 Bergen, Norway
[2]School of Geosciences, University of Aberdeen, Meston Building, Kings College, Aberdeen AB24 3UE, UK
[3]OMT Tech AS, Norway

*Correspondence to*: Simon J. Buckley (simon.buckley@norceresearch.no)

**Abstract.** V3Geo is a cloud-based repository for publishing virtual 3D models in geoscience. The system allows storage,
search and visualisation of models typically acquired using techniques such as photogrammetry and laser scanning. The
platform has been developed to handle models at the range of scales typically used by geoscientists, from microscopic, hand
samples and fossils through to outcrop sections covering metres to tens of kilometres. The cloud storage system serves the
models to a purpose-built 3D web viewer. Models are tiled to ensure efficient streaming over the internet. The web viewer
allows 3D models to be interactively explored without the need for specialist software to be installed. A measurement tool
enables users to measure simple dimensions, such as widths, thicknesses, fault throws and more. V3Geo allows very large
models comprising multiple sections and is designed to include additional interpretation layers. The specific focus on
geoscience data is supported by defined metadata and a classification schema. Public and private storage are available, and
public models are assigned Creative Commons licenses to govern content usage. This paper presents V3Geo as a sustainable
resource for the geoscience community, including the motivation, main characteristics, and features. Example usage scenarios
are highlighted: from undergraduate geology teaching, supporting virtual geoscience education, and preparing virtual field
trips based on V3Geo models. Finally, best practise guidelines for preparing 3D model contributions for publication on V3Geo
are included as an Appendix.

## 1 Introduction

Virtual 3D models, including virtual outcrops, are an increasingly standard data type underpinning many applications in
geology and the wider geoscience discipline. These 3D models, acquired using photogrammetry (frequently referred to as
structure-from-motion; SfM), laser scanning (lidar) or other optical 3D modelling methods, offer high resolution and accurate
representations of real-world topography, hand samples, fossils and more, for a range of quantitative and qualitative purposes.
Examples are detailed interpretations and measurements of statistically significant geometric data for analysis of different
geological settings (e.g. Enge et al., 2010; Eide et al., 2014; Rittersbacher et al., 2014), supporting teaching (Senger et al.,
2020; Bond and Cawood, 2021; Senger et al., 2021), through to being a fundamental component of virtual field trips (e.g.



Klippel et al., 2019). The adoption of 3D acquisition techniques in geoscience is now widespread and universal, driven by rapid technological advances in automated image matching available in relatively low cost photogrammetric software, unmanned aerial vehicles (drones) as imaging platforms, and the ubiquity of digital cameras and fast computing hardware for acquiring and processing large 3D datasets (Eltner et al., 2016; Granshaw, 2018). The COVID-19 pandemic has intensified

adoption and served to shift focus in the geoscience community from acquiring suitable 3D models, to more effectively utilising these models within the range of research, educational and professional scenarios envisaged by earlier authors (e.g. McCaffrey et al., 2005; Buckley et al., 2008a; Pavlis and Mason, 2017).

Despite the increased ease of model acquisition, a major challenge associated with virtual 3D models has been the large datasets
and heavy graphics hardware requirements needed for visualisation and analysis. Typical 3D datasets may comprise hundreds or thousands of digital images (for photogrammetry), or tens of individual scans (for lidar). While processed point clouds can be used for further analysis, it is common to generate a triangulated surface mesh with photo-texture, to give a continuous representation of the field area or sample at the resolution of the original camera imagery (Buckley et al., 2008a). Although a textured model facilitates interpretation and accessibility, structured as a standard 3D graphics file format (e.g. .obj) they
typically still consist of millions of triangles and associated texture images, resulting in gigabytes of data when loaded in graphics memory. Consequently, higher end computer hardware and graphics cards have traditionally been used to work with these models, precluding sharing, particularly over the internet where such monolithic data organisation is highly unoptimized for dynamic streaming. Tiled 3D models, where data is organised into spatially coherent sections and multiple levels of detail (LODs; e.g. Borgeat et al., 2005), allow 3D viewing software to load individual tiles on demand (Buckley et al., 2008b). The
original 3D model is processed to create a hierarchy of files on disk, which are then loaded according to the user's position while navigating in the 3D scene. In this way, geologists can seamlessly move between overview and high-resolution close-up views of the geology, potentially over many kilometres of exposure or between multiple localities within the same viewing session.

Tiled models and out of core rendering (Buckley et al., 2019) circumvent the performance issues associated with conventional file formats. However, transferring such datasets (potentially many gigabytes and thousands of files) for sharing purposes can still be inefficient. Current cloud storage and web visualisation technology give new opportunities for effectively disseminating virtual 3D models, as storing tiled models in the cloud "hides" the cost of transferring large datasets between users, with only the required model tiles accessed on demand through an online database and file repository.

Web-based solutions for viewing 3D models have been proposed previously (e.g. Sawicki and Chaber, 2013) and are today available as standalone commercial solutions (e.g. Sketchfab, 2021), or implemented on top of existing open source or commercial framework libraries. Application programming interfaces (APIs) such as WebGL allow programmers to create graphical applications in the web browser across multiple platforms and devices without requiring software installation, which



can be a potential barrier to adoption. Within geology and geoscience, Cawood and Bond (2019) have created e-Rock as an online 3D model repository, based on embedding the underlying Sketchfab viewer. The SAFARI project (https://safaridb.com) has long used a web 3D viewer to display virtual outcrop models for participating companies and collaborating research groups (Howell et al., 2014). Nesbit et al. (2020) presented case studies utilising several web-based solutions including Sketchfab, an open source point cloud viewer and the Unity gaming engine, assessing each based on factors such as size of dataset handled,

ease of use and level of programming knowledge required.

Although these studies highlight the practical benefits of web-based sharing of 3D models in geoscience, no current single repository has been presented for scientific and professional purposes. Solutions are limited in file (and therefore dataset) size, precluding many of the details needed for interpretation, do not allow supplementary interpretations or datasets, or are too

broad in scope, covering all areas of society rather than being tailored to the geoscience community. Importantly, quality control is not considered, but is increasingly important given the current move towards open data and FAIR (Findability, Accessibility, Interoperability and Reusability; Wilkinson et al., 2016) principles being enforced by governments and research councils as the custodians of publicly funded science. This has led to focused databases, applications and services growing to serve the geoscience community (e.g. StraboSpot; Walker et al., 2019).


Increased adoption of virtual 3D models in geoscience, brought about by ease of acquisition and a raised awareness of their contribution in analysis and education, affords new opportunities when combined with the potential of web technologies. With 3D methods being increasingly used in fieldwork and within the laboratory, a greater number of field areas and specimens are being acquired in 3D. With a common sharing platform, this can avoid repeating acquisition efforts through routine sharing of

high quality and reliable models.

This paper presents V3Geo (Virtual 3D Geoscience) as a novel repository for storing and sharing digital 3D models in geoscience. The main contribution is to provide a sustainable resource that is tailored to the needs of the scientific and professional community, in terms of scope, quality and reliability of content, level of model detail, performance and with a

future perspective to address the rapidly changing technological landscape in virtual geoscience. The paper gives a high-level description of the main implementation considerations and functionality. Next, key considerations to create high quality 3D models are outlined, together with the technical details for preparing contributions to populate the V3Geo repository (detailed considerations for preparing model contributions are given as an Appendix with the aim of aiding new users and increasing the uptake of 3D modelling in geoscience). Finally, four usage scenarios illustrate how the V3Geo repository is being used as

a resource for teaching geological field concepts, supporting teaching of digital field methods in virtual geoscience, as well as preparing multi-locality virtual field trips (VFTs). It is worth stressing at the outset that as with many technological developments digitising real-world field areas, the purpose of V3Geo is in no way meant to replace actual fieldwork and excursions. Instead, V3Geo provides a platform for sharing high quality 3D data, for increasing the accessibility and range of



field areas available to a global community, complementing physical field geoscience and education, and preserving field sites

for posterity (where physical changes, political or access restrictions may preclude revisiting localities).

## 2 V3Geo repository overview

### 2.1 Specification

V3Geo was conceived to simplify access and sharing of virtual 3D models in geology and geoscience. Due to the large 3D datasets, publishing models at their original resolution, sharing results with collaborators, project sponsors, or distributing a

dataset to support a university class has been a major barrier to acceptance of virtual methods. Consequently, the initial specification for V3Geo was devised to meet the following key criteria:

- Support large 3D models with multiple model sections. Standard solutions allow a single model upload, which precludes the ability to combine different sections, such as representing multiple faces of an outcrop, or a continuous exposure that stretches over a large distance and requires splitting into multiple chunks to facilitate processing.

- Allow high resolution datasets to be stored and accessed. Typically, previous sharing solutions compromise resolution to meet the requirements of a single model within a specified file size. With V3Geo, we aim to allow the original resolution dataset to be available, requiring tiled models to be supported.

- Have a simple classification schema and metadata, allowing the database of models to be easily searched and filtered according to defined characteristics, geographical location etc.

- Ensure stored 3D models are viewable over the web without software or plugin installation, and compatible with a wide range of platforms and device hardware.

- Respect data ownership, copyright, and usage.

- Safeguard model data storage, user data and access using data governance and security standards.

- The system should be scalable to adapt to many concurrent users independent of geographical area.

- Designed to allow for future functionality, usage scenarios and technology trends.

### 2.2 V3Geo architecture and framework

The V3Geo repository is implemented as a PostgreSQL database hosted on Amazon Web Services (AWS). PostgreSQL (https://www.postgresql.org) is a well-established, open source relational database system, and is used to store user profiles and model metadata as tables, which are queried using the structured query language (SQL). The virtual 3D models, images

and other static files are stored using the AWS Simple Cloud Storage (S3) accessed through the CloudFront content delivery network (CDN). A GraphQL API is used to interact with the underlying database. This was chosen over more traditional REST APIs for a number of reasons: to limit the number of API endpoints to implement and maintain, to separate the API from the needs of specific client applications, and to minimise the number of API calls needed to retrieve data from the server. From



our experience, use of a graph approach reduces the burden on the server-side, allowing client applications to tailor their

queries and data usage according to actual needs, without having to specifically alter the API endpoints. User authentication and security is performed by the external Auth0 service, which is in turn used to manage the authorisation within the V3Geo database through the GraphQL API. This allows user profiles to be established and single sign-on (SSO) to be used to manage user identities, data protection and roles within V3Geo. Finally, two client web applications provide the V3Geo front end for users. The main website application (https://v3geo.com) is the primary front end to browse, view and manage model data.

Virtual 3D models are visualised on demand using a client web viewer, which is based on WebGL for 3D graphics.

### 2.3 Main V3Geo functionality

This section describes the main user-facing characteristics of the V3Geo system at the time of writing.

#### 2.3.1 Roles

V3Geo allows several user roles within the system. Standard (anonymous) website users can browse the website, search for

available 3D models, view a model's metadata, and launch the web viewer to explore the stored 3D model. Standard registered users can create public model pages (models that are visible to all users) and submit them for inclusion in the database. Private users can, in addition, create model pages that are only visible to themselves by default. Finally, a reviewer role can be granted to specific users to facilitate the technical quality control process of approving public model contributions into the repository. All users wishing to contribute or review content in V3Geo must make a user account (or associate an existing SSO account)

and login to access the content management and review tools.

#### 2.3.2 Model licenses

Copyright and data ownership of contributed models is retained by the author, with usage governed by Creative Commons (CC) licenses (Creative Commons, 2021). Before a public model is published, the author must select one of either CC BY 4.0 or CC BY-NC 4.0 (for non-commercial use) or CC0 (copyright assigned to public domain). These licenses require that the

name of the model entry, the author and the model source are cited in any usage of the dataset. Private models do not require a license unless the owner later wishes to make the model public, in which case a CC license must be assigned. By allowing this selection of licenses, we aim to create a balance between the accreditation and copyright needs of authors, guidelines for open data being introduced by funding agencies, as well as ensuring use by the professional geoscience community.

#### 2.3.3 Metadata and search interface

Models within V3Geo are assigned simple metadata and tags to enable them to be filtered and searched. Basic information includes the model's name (typically the name of a specific field area, sample etc), author, funding source, license, country/region, a short description of what the model represents, and citations to relevant literature. The model description and literature fields can be formatted with a Markdown text editor to allow clickable links and aesthetic content. In addition, models





are classified according to pre-defined fields representing size of the modelled area, and smallest visible feature (equivalent to
ground sample distance; GSD; and a subjective indication of the geological detail visible within the 3D model, and therefore
the model's resolution). The metadata also includes georeferencing information, currently an approximate placement of the
model centre on an interactive map, dictating the position in world coordinates. Authors can also add several screenshots of
the model, which will play in a carousel on the published model page. Finally, the model is tagged with one or more entries
from lists of geology type (main geology category and sub-category) and geological age. These lists are currently predefined,
though can be easily expanded in the future to cover the wider geoscience.

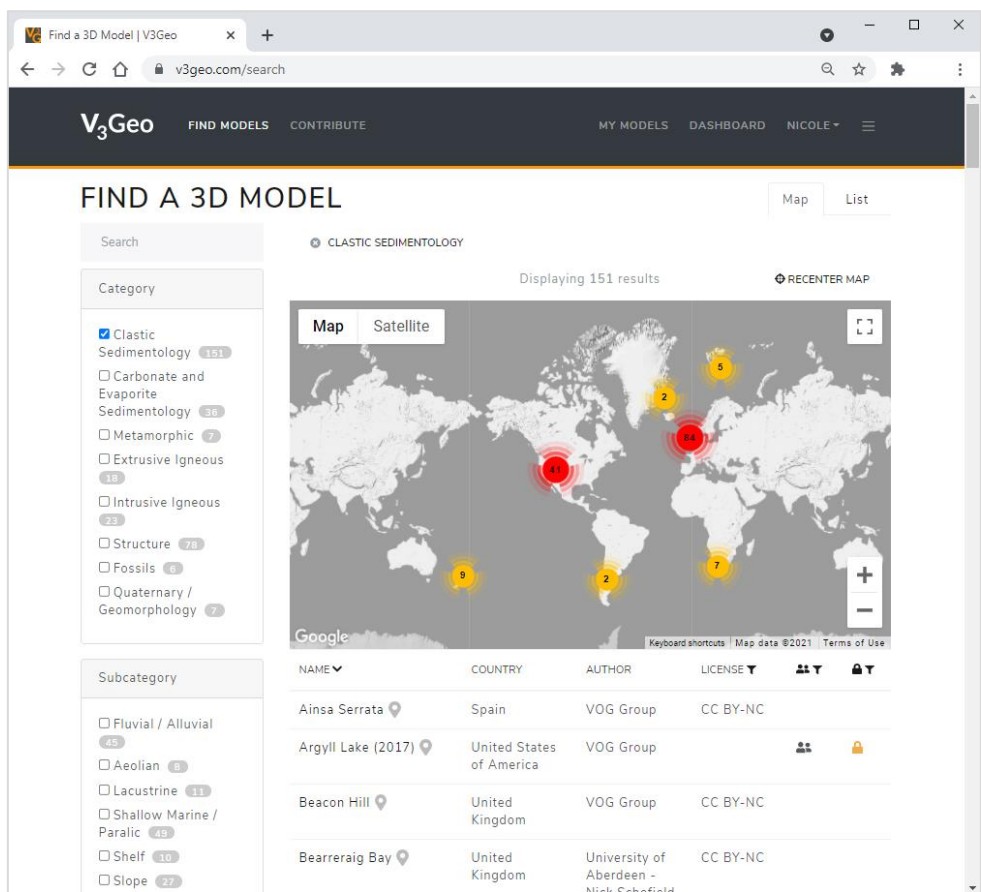

Figure 1. Search interface in V3Geo web application, showing results when filtering the content by Clastic Sedimentology.

Published models can be browsed using the Find Virtual 3D Model page on the V3Geo web application (Figure 1), which
gives an overview of all models that a particular user is authorized to access. Filters allow users to isolate available models by
geology type, geological age and country, and all results are displayed as pins on a map, a table with sortable columns (by
name, country, author, license type, visibility and sharing status), as well as in a list of model descriptions. A free text search
is also available, which can be combined with the filters to make a customized search of the database. In all cases, the map



updates interactively according to the current search results; users can click on highlighted pins or the entries in the returned
list to display the model page with images, description and metadata (Figure 2).

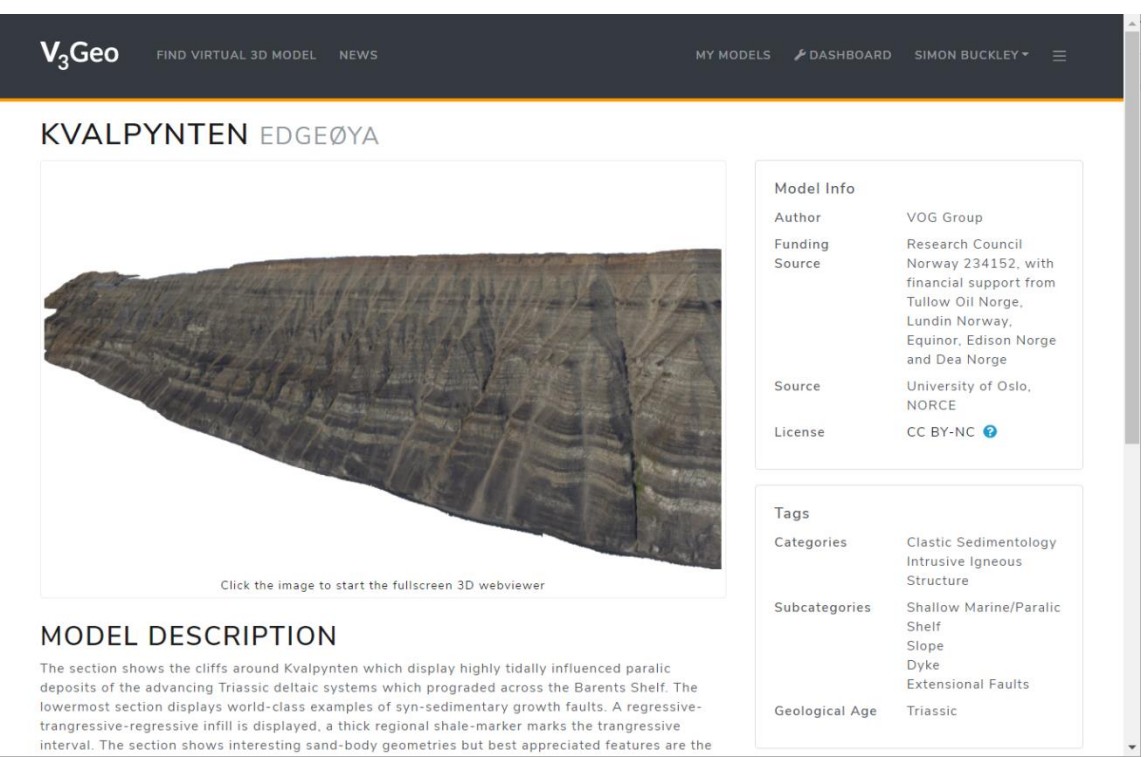

**Figure 2. Model page on V3Geo, showing metadata, model description, tags and image carousel serving as a hyperlink to launch the**
**3D model viewer. Example shown is from Kvalpynten, Edgeøya, Svalbard (Anell et al., 2021).**

### 2.3.4 V3Geo web viewer

The model description page includes a link to visualise the actual 3D model data using the V3Geo web viewer. This opens a
new browser window with the model centred in the 3D view, and a simple GUI with options for setting the vertical exaggeration
of the 3D scene, level of detail of loaded data (adjustable according to the internet connection and computing hardware; the
default can be accepted in most situations), and adjusting the brightness and contrast of the model texture (Figure 3). The latter
is useful in areas where lighting conditions were not optimal during image acquisition, such as on outcrop faces with strong
shadows contrasted with bright sunlight in other parts of the model. The brightness and contrast can be temporarily adjusted
during navigation to allow more details to become visible. A tool is available for measuring distances between two or more
points placed by user.




As the user zooms in, the hierarchy of model tiles is successively traversed according to the current position and orientation of the view relative to the model itself, and the level of detail loaded accordingly. Using this approach, models comprising multiple sections, large areas and many data files can be seamlessly interacted with over a reasonable internet connection.

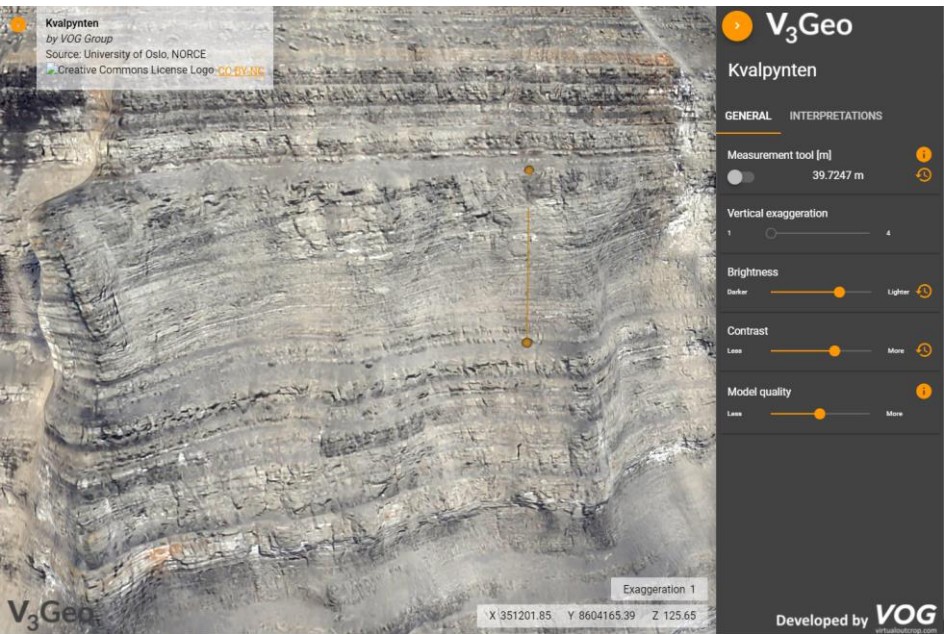


**Figure 3. V3Geo web viewer for interactively navigating around tiled 3D models stored in the repository. Example shows point bars exposed at Kvalpynten (Anell et al., 2021). Complete model covers c. 26 km of outcrop, combined from eight processed sections. Measurement tool has been used to mark a distance of 39.725 m.**

In addition to being integrated in the web application, the V3Geo web viewer is also embeddable in other web pages. This
makes it possible to utilise V3Geo content for many scenarios, such as custom sites covering a particular type of geology, an online virtual locality overview or excursion stop, or teaching material or exercises for a university course module. In HTML code, the web viewer is embedded as an iframe container using the following basic syntax:

```
<iframe id="viewer" src="https://v3geo.com/viewer/index.html#/1" width="700" height="300">
</iframe>
```

This example will embed database model number 1 (Ainsa Quarry) with a frame size of $700 \times 300$ pixels. On smaller screens and embedded window sizes, the GUI will be hidden by default.



### 2.3.5 Connection to LIME through V3Geo model loader

The V3Geo web application interacts with the secured database and cloud storage through the GraphQL API. The same API will in the future allow other client applications to access the data in V3Geo, though it has not yet been published due to the frequent changes in API endpoints as the system develops rapidly (see Section 5.1). As the authors also develop the LIME desktop software for interpretation, visualisation, and communication of 3D models in geoscience (Buckley et al., 2019), we have been able to integrate the fledgling V3Geo API to load cloud models. LIME has been modified to handle V3Geo user

authentication, as well as adding a browse dialog allowing users to filter the list of models in a similar way as the V3Geo web interface. Users can select models, which are then loaded into the LIME environment and can be interacted with like any other 3D model stored conventionally on local disk storage. LIME has previously been developed to work with tiled 3D models. Once loaded, models can be interpreted using lines, orientation planes, overlays and points of interest, as well as complemented with additional datasets such as GIS data, image panels representing cross sections, field photos, geophysical sections and

more (Buckley et al., 2019). Therefore, loading models from the V3Geo cloud is a powerful approach for sharing datasets and projects between collaborators, performing detailed interpretation, and working with multiple models and localities within the same environment, beyond what is possible in the V3Geo 3D web viewer. The key benefit of sharing projects in this way is that the 3D models are accessed from the cloud and only the "additional data" that augments the project need to be shared. Given that a project may contain several gigabytes of models but only a few tens of megabytes of supplementary data, it is

much easier to share with collaborators, students and virtual field trip participants (Section 4.3).

### 3 Guidelines for contributing models to V3Geo

### 3.1 Quality control is aided by technical "review"

An important advantage of V3Geo is the ability to store and access large 3D models with multiple sections, allowing wide areas and high-resolution data to be incorporated in the repository. In addition, we believe that a technical quality control

ensures a minimum standard to the models in the database so that they can be relied upon in scientific and professional use, and thus furthers the community expertise level and adoption of virtual 3D models in geology and geoscience. This reflects the general aim to ensure rigorousness in 3D modelling approaches in photogrammetry and laser scanning to ensure 1) reliability of results and error reporting in scientific publications based on the 3D models; 2) a general reliability when these models themselves are made available for use by others. It is out of the scope of this article to discuss potential error sources

in detail; readers are instead referred to earlier work by e.g. Chandler (1999), Buckley et al. (2008a), Eltner et al. (2016) and James et al. (2019) for reflections on issues related to data quality.

New public submissions to V3Geo are therefore subject to a basic technical "review", so that they meet fundamental requirements related to suitability of content, georeferencing and scaling, 3D mesh and texture size, and model formatting.





This practice has long been built up through workflow developments and internal quality control by the authors, and our experience of interacting with potential contributors, gathered since the initial release of V3Geo in April 2020, shows that there is value for the wider community in putting in place an approach to ensure reliability. It is important to note that the technical check is only to ensure the 3D models themselves meet a minimum standard, it is not a traditional scientific review, such as applied to a journal contribution, and any metadata descriptions that include observations or interpretations, or

references to further work, are the responsibility of the author to scrutinise prior to submission. It is instead analogous to an initial assessment by a journal's editorial team to ensure a submission meets the published guidelines. The V3Geo technical review is currently performed by members of the author group, though we plan to establish an "editorial board" of technically savvy geoscientists who have experience with 3D modelling methods and applications.

To facilitate high quality contributions to V3Geo that successfully meet the technical requirements (and therefore a fast turnaround time), we have prepared guidelines for model preparation, published for the benefit of the geoscience community (Appendix A and in Howell et al., 2021). Although prepared with V3Geo in mind, they are equally applicable to more general 3D modelling and viewing software. The provided workflows document an approach to 3D model acquisition and processing we have found to be successful for many applications, but are not the only way to reach high quality models. Rather, they

provide one route that we hope is useful to new users of 3D models in geoscience, as well as offering some guidance for more experienced users who wish to contribute models to V3Geo at a level of detail not supported by contemporary alternatives to sharing.

**3.2 General procedure**

V3Geo expects two main parts to be prepared for any contribution: 1) 3D model data files; 2) metadata describing the model

and characteristics of the field area, hand sample etc. A prospective author to V3Geo should register with the repository using the web application at https://v3geo.com. Here, it is possible to create a new model page and decide whether a model has public or private scope. This document covers public contributions with technical review. The model page is populated with the required fields, CC license, image(s), georeference and tags covered in Section 2.2.3. In addition, a file dropzone allows model sections to be uploaded (can be several per published model) as zipped files (according to Appendix A). Once

completed, a preview of the model page can be checked before it is submitted for technical review. After submission, a simple review process is started, where the review team is notified, the 3D model is inspected according to the guidelines (Appendix A). The technical check will result in either 1) approval of the model and publication to V3Geo; or 2) specific feedback for the author to improve the model and invitation to resubmit. Examples of common feedback are provided in Table 1.

**Table 1. Common feedback on 3D model contributions following V3Geo technical check.**

| *Category* | *Common issues* |
| --- | --- |





| Georeferencing | Model incorrectly positioned in world space when compared to e.g. map or Google Earth and versus supplied metadata |
|---|---|
| | Model coordinates incompatible (i.e. exported in latitude/longitude or geocentric) instead of projected (Cartesian) |
| | Model not orientated or scaled correctly |
| Registration errors | Dataset has gross modelling errors, resulting in warping, distortion, or double surfaces |
| Resolution and optimisation | Mesh surface is noisy, has holes, or modelling artefacts |
| | Individual model sections are too large for efficient LOD generation – split up model into chunks of c. 3m triangles |
| | Texture has been generated a too low resolution – increase the resolution or number of texture images |

Based on our experience, the first model submission from a particular author may result in issues with the data quality being highlighted and suggestions on how to address these sent to the author for consideration. Later submissions are then prepared according to the guidelines and approved. Hence, by publishing this guideline we hope to aid new users and contribute to an
increased standard of 3D model quality in the geoscience community.

**3.3 Preparation and submission of 3D model files**

The main requirements for preparing 3D model files for submission are related to georeferencing, ensuring a balanced resolution of model mesh and texture per input model section, and the final export format. Considerations and guidelines for preparing and formatting a model for V3Geo submission are documented in Appendix A.


The result of 3D model processing and formatting is one standard OBJ file per model section, which is then ready for submission to V3Geo. At the time of writing (and see Section 5.1, COVID-19 perspective), model files are uploaded as part of the V3Geo model submission process but must be converted by the project team before publication. Submitted model files are checked and, once they meet the technical guidelines, are converted into optimised tiled models for uploading to V3Geo.
In this step, an automatic algorithm (based on the initial steps proposed by Buckley et al., 2008b) converts each OBJ file into a hierarchy of LOD tiles suitable for streaming over the internet. This tiled model is uploaded to the V3Geo storage before the model itself is approved for publication and becomes visible to general site users.

**4 V3Geo example usage scenarios**

Since its inception, the V3Geo repository has been receiving contributions and has been utilised by many users in the
international scientific and professional geoscience community around the world, in particular as an educational resource to





aid field-based teaching during the COVID-19 pandemic. In this section, we detail cases from our own activities where V3Geo has contributed.

## 4.1 Publications based on 3D models

As there is increasing utilisation of the 3D geoscience approach, especially virtual outcrops, there is a growing body of
scientific literature being published based on virtual outcrop data. At present the main method to include the 3D model is through static screen grabs, which are typically unsatisfactory, given the fundamental benefit conferred by viewing field localities from different perspectives in a 3D viewer. Our goal is that V3Geo will become a standard repository for virtual outcrops and 3D geoscience models linked to scientific publication. At present V3Geo includes a number of virtual outcrops which have been used in scientific publications, such as Beckwith Plateau, (https://v3geo.com/model/101; Eide et al. 2014),
Panther Tongue (https://v3geo.com/model/42; Enge et al. 2010), Kvalpynten (https://v3geo.com/model/90; Anell et al., 2021) and numerous Yorkshire Coast outcrops (Rahman, 2019). This number is expected to grow significantly in the coming years.

## 4.2 Classroom teaching

### 4.2.1 Geology classes

Virtual outcrops can be used to augment both the taught and practical aspects of traditional classroom courses. As an example,
V3Geo has been used extensively in the Applied Sedimentology course (GL5044) at University of Aberdeen (UoA). GL5044 forms part of the Integrated Petroleum Geoscience (IPG) MSc programme. The course covers all aspects of clastic sedimentology with special focus on how various sedimentary bodies (architectural elements, *sensu* Miall, 1985) control fluid flow in the subsurface. V3Geo contains virtual outcrops representing all the principal depositional environments (Table 2) and provides a unique opportunity for the course leader to discuss key elements and display their 3D properties during the course.
The class is then set an exercise to prepare "an atlas of sedimentary architecture". For this they use V3Geo and other published resources to document the diagnostic properties, geometries and architectures of 50 elements that are potentially relevant in subsurface studies. This encourages the students to interact with V3Geo and to explore the content of the database in their own time, to identify elements, describe them and make measurements. Student feedback has been very positive, with 86% saying that it was a useful exercise that improved understanding of sedimentary systems and 92% saying that V3Geo was a good way
to study sedimentary rocks in the classroom. The students enjoyed interacting with the database and found the organisation, layout and user interface easy to navigate and use.

**Table 2. V3Geo virtual outcrop models used for teaching sedimentary environments at UoA.**

| *Depositional environment* | *Outcrop name* | *V3Geo model* |
|---|---|---|
| Alluvial fan | Aguero, Spain | https://v3geo.com/model/14 |
| Braided fluvial | Castlegate, Utah, USA | https://v3geo.com/model/86 |





| Meandering fluvial | Cinca Canal, Spain | https://v3geo.com/model/12 |
|---|---|---|
| Overbank | Whitby Beach, UK | https://v3geo.com/model/60 |
| Aeolian dunes | Huab Valley – Big Dune, Namibia | https://v3geo.com/model/221 |
| Lacustrine | Argyll Lake, Utah, USA | https://v3geo.com/model/96 |
| Wave-dominated shoreline | Blue Castle Canyon, Utah, USA | https://v3geo.com/model/97 |
| Tidally-dominated shoreline | Elgol Beach, Skye, UK | https://v3geo.com/model/47 |
| River-dominated delta | Ivie Creek, Utah, USA | https://v3geo.com/model/52 |
| Estuary | Thompson Canyon, Utah, USA | https://v3geo.com/model/9 |
| Shelf turbidites | Hatch Mesa, Utah, USA | https://v3geo.com/model/78 |
| Slope channels | Ainsa Quarry, Spain | https://v3geo.com/model/30 |
| Basin floor fan | Tanqua Karoo, South Africa | https://v3geo.com/model/13 |
| Mass transport complex | El Gordo Megabed, Spain | https://v3geo.com/model/4 |

V3Geo was also used in a UoA first-year course that introduces the fundamentals of geology (GL1005 – The nature of the environment through geological time). Students were required to study a variety of models in V3Geo that illustrate important relationships such as intrusions, faults, and folds, and they were encouraged to sketch relationships between the different rock units, measure key elements and provide digital or hand drawn diagrams. The work formed part of the final assessment for the course. This exercise comes before they have ever been to the field and provides an opportunity to study relationships in a way
that would typically not happen until their first field trip at the end of their first year. Again, feedback was positive, with the students appreciating being able to interact and move around the V3Geo models.

**4.2.2 Concepts in virtual geoscience**

The University of Bergen runs an intensive module for final year geoscience MSc students in the autumn semester (GEOV364), with one week devoted to spatial acquisition techniques and their potential in geology and geoscience. Students learn about
the basics of lidar and photogrammetric acquisition and processing, GIS and how to visualise, interpret and analyse 3D models. Photogrammetry is major enabler for geoscientists (e.g. James et al., 2019), which is covered on the second day of the course. After an introductory lecture, students acquire their own image sets for photogrammetric processing using objects around the campus, such as building facades, walls, hand samples etc. They quickly learn what makes a successful subject and dataset configuration for photogrammetric processing. The software used is VisualSFM (Wu, 2013) which, while no longer the most
up to date, has the advantage of being freely available and is interactive to show the visual progress of the 3D image reconstruction and sparse point cloud generation, in what is otherwise a "black box".





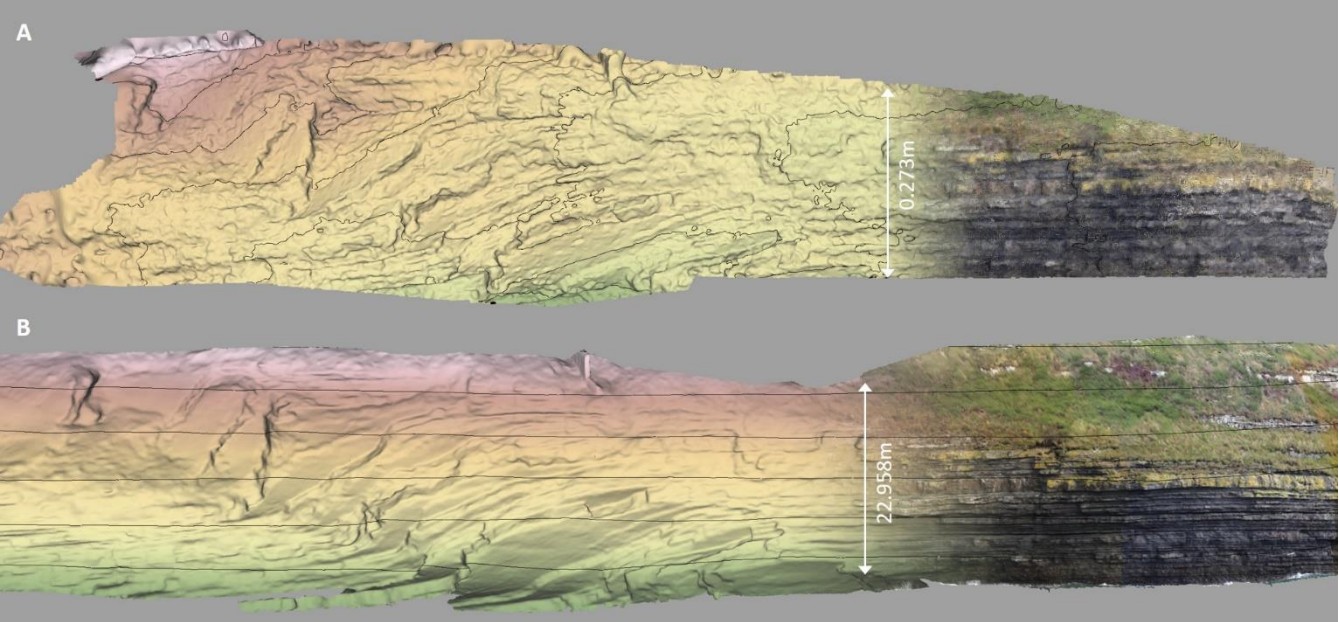

**Figure 4. Virtual outcrop model of Kilcredaun, Ireland (Kilcredaun, VOG Group, https://v3geo.com/model/67). A) 3D model**
**processed without georeferencing by students during classroom exercise: scale and orientation of model is incorrect (0.05 m**
**contours); B) Final model in V3Geo (5 m contours). Models coloured by elevation (white=high through to green=low).**

After working up their own datasets to the dense point cloud stage (with discussion and feedback on the suitability of their

chosen object and imaging strategy), the students are given a pre-acquired image set of an outcrop available in V3Geo (a subset

and resampled slightly to make the processing time appropriate to the length of session). They extract a dense 3D point cloud,

which is then loaded into CloudCompare (https://www.cloudcompare.org) for measurement and meshing. As no

georeferencing information is given, the students are asked to measure the outcrop height and use the default viewing directions

(e.g. map view), and discuss reasons for the point cloud scale and orientation not making sense (they typically find that the

axes of the model are swapped so the outcrop face is "up", and that the height of the outcrop is a several centimetres, as a result

of the photogrammetric initialisation using an arbitrary coordinate system; Figure 4). They are then informed about the final,

published model on V3Geo and asked to inspect and measure the outcrop in the web viewer. Finally, they use the V3Geo

model connection in LIME to load the same model and perform a coarse georeferencing to the V3Geo model, so they can

compare the relative quality of the two datasets. This exercise, taking around three hours, gives understanding of some of the

pitfalls in image acquisition, texture requirements, and the importance of georeferencing and scaling. Having the final V3Geo

model accessible and published is a way to communicate the power of the photogrammetric technique and its real-world

application for geoscientists.





### 4.3 Virtual field trips

One of the most obvious applications for virtual outcrops is as the basis for virtual field trips. There are several ways that V3Geo can be used to build VFTs. The simplest is to use the V3Geo web viewer to direct students to the outcrops and encourage them to make observations and measurements in much the same way that they would in the field. A simple example of this was during the COVID-19 pandemic, when field teaching was not possible, University of Aberdeen 4th year undergraduate students used the St Cyrus model (https://v3geo.com/model/120) as a replacement for an existing field trip to the same location. Course exercises were modified to work from the V3Geo model, and it was possible for students to observe and map the sediment/lava interaction in the Lower Devonian sections that are well exposed in the cliffs behind the bay. A similar use was made of the Ainsa Quarry model (https://v3geo.com/model/30) by Imperial College London students on a VFT to the Pyrenees. As the V3Geo viewer can be embedded in external websites, it is also possible to incorporate multiple models within online virtual field trips, allowing participants to view and learn from the various localities in the context of a more extended virtual field guide. Models can also be combined with Google Earth to provide regional context, while the V3Geo models give detailed views of an outcrop locality.

At UoA, the most widely used method for delivery of VFTs using data from V3Geo is through the connection to LIME. This allowed us to build more complex and interactive VFTs for the students. Pugsley et al. (this volume) provide a detailed overview of the VFT creation methodology and teaching methods, as well as a review of the outcomes of running VFTs at the University of Aberdeen. This work is based on two VFTs run for MSc students – an 11-day trip to Utah and a 5-day trip to the Spanish Pyrenees. Both trips were run almost exclusively in LIME and made extensive use of models from V3Geo (Figure 5). The Utah VFT used a total of 48 virtual outcrop models, 21 from V3Geo, all augmented with other geological data. The students ran LIME locally or via a virtual machine hosted by the university, and they downloaded the project files required for the day each morning. Using V3Geo as the repository for the 3D models kept the project files to a minimum size and made it possible for participants with data download restrictions to participate effectively.



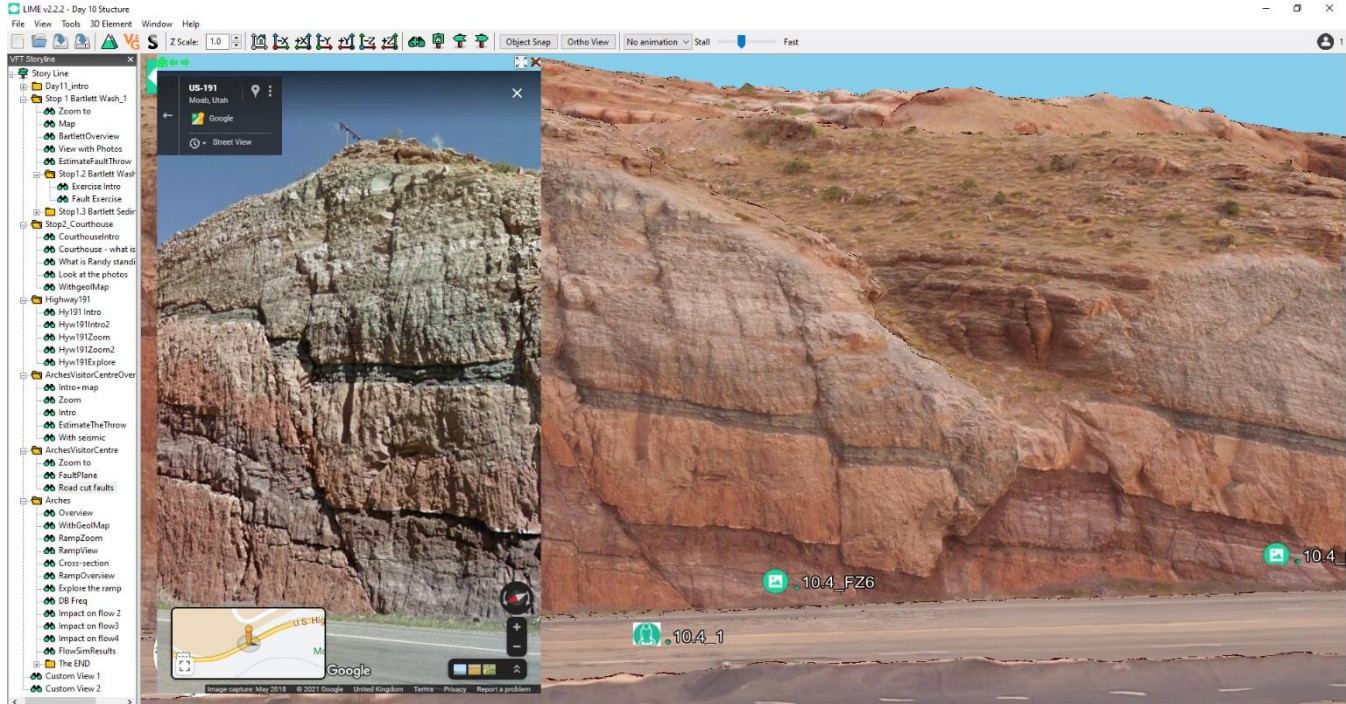

**Figure 5. Arches Road Cut, Utah, USA (https://v3geo.com/model/98), one of the stops on the University of Aberdeen MSc virtual field trip to Utah. Multiple days of VFT content are prepared in LIME using multiple V3Geo models and additional material including photos, 360° panorama images, logs and other field data types, and shared with students during the course.**

# 5 Future directions for V3Geo

## 5.1 COVID-19 perspective

The first release of V3Geo was made in April 2020, with development fast-tracked in response to the first wave of COVID-19 lockdowns affecting Europe and much of the world from March 2020. Although parts of the system were complete at this point, it was not planned to make the public release of V3Geo until late 2020, when more functionality for registered users had been completed. However, restrictions on travel and in-person teaching in educational organisations resulted in cancellation of conventional field-based teaching and excursions, requiring alternatives to be quickly established. The launch of the browse and viewing functionality of V3Geo was therefore made to help educators and the geoscience community to weather the pandemic, even it was not complete according to the original project vision. V3Geo continues to evolve, with a roadmap for future direction in place to ensure the repository develops as a resource for publishing 3D data for the scientific and professional community. The following section outlines some areas of development prioritised in the shorter term.



## 5.2 V3Geo is interpretation-ready

V3Geo currently allows storage and viewing of high-resolution 3D models representing the surface state at the time of capture. However, the repository has been developed to incorporate additional interpretations created using the 3D models, such as

marking key stratigraphy, structure, geomorphological or geological features, etc. Authors will be able to submit their interpretations together with a model, or potentially create new interpretations based on another published model within the database. Interpretations are planned as overlay layers on top of the 3D model data (e.g. Buckley et al., 2019) and will have access individually governed by similar system roles as for the model data. The aim is to facilitate publication of interpreted models made within the scope of a student project or research article etc, allowing open access to datasets that have traditionally

been difficult to share in 3D without resorting to 2D screenshots or interpretation panels that are non-interactive, and therefore difficult to appreciate and reproduce by publication readers.

## 5.3 Facilitating 3D publication

Several measures will assist authors to publish 3D models on V3Geo. It is planned to introduce an automatic converter to facilitate the process of integrating the 3D model files into the V3Geo database. Currently this is performed by the project

team following submission by authors, using the tiled model converter (Section 3.3) and upload API. In the future this step will be deployable on the V3Geo cloud infrastructure, to reduce time from submission to publication. Some further optimisation is being carried out to ensure robustness to the many different 3D model cases and data qualities that are generated – aided by adopting model preparation approaches such as those advocated in Appendix A – before deployment on the server is completed.


File formats for 3D graphics on the web are not yet fully standardised between processing applications and in-browser rendering, particularly for tiled models. As one of the most widely used 3D exchange formats in graphics software, the OBJ format is currently favoured for submission to V3Geo, even if it is not efficient in terms of rendering and streaming. Internally, the V3Geo tiled models are currently stored in a JSON format with supplementary binary geometry arrays and JPG image

textures. However, a number of tiled model formats are now gaining traction for representing large, real-world 3D environments required for applications such as V3Geo. Examples are the Open Geospatial Consortium (OGC) Community standards for 3DTiles (https://www.ogc.org/standards/3DTiles) and i3S/Scene Layer Package (SLPK; http://docs.opengeospatial.org/cs/17-014r5/17-014r5.html#89) formats that have the potential to address issues around interoperability and ensuring standard coordinate reference systems are enforced for field-based datasets.


Publication of data is becoming increasingly recognised in science, with open data initiatives being established to govern the release of public-funded datasets. Funding agencies require data management plans in grant applications, and permissive licenses for data are becoming standard, even though making large 3D datasets has conventionally been problematic.





Publishing 3D datasets to V3Geo will conform to the drive for open data access, allowing authors to use V3Geo as a repository
for public 3D models, as well as a landing page for supplementary 3D datasets supporting a journal article. Although
acceptance and author credit for data publication is normalising, we plan to introduce the option for supporting Digital Object
Identifiers (DOIs) in V3Geo, so that the repository conforms more closely to FAIR principles.

## 5.4 Sustainability

An important consideration for V3Geo, as for other community-driven initiatives realising infrastructure, databases or
software, is the future development and maintenance needed to create a sustainable and long-term resource. This is especially
pertinent for systems based on rapidly changing web and 3D technology, where innovations and standards are in continuous
evolution. For V3Geo, large dataset storage and traffic resulting from high engagement from users, as well as maintenance
and development by skilled engineers needed to work on a complex interaction of components, requires an associated model
for long-term operation. While this is not yet determined, we are engaged with public agencies, learned societies and even
private actors to support and evolve the repository. In the shorter-term, a subscription approach for private and organisation
storage is one potential means to ensure the system's durability and public accessibility for the scientific community.

## Conclusions

This paper has presented V3Geo as a web repository for sharing and browsing virtual 3D geoscience models. The main goal
is to allow the geoscience community to find 3D models to include in educational, scientific and professional activities, as well
as acting as a repository and archive for sharing and publishing 3D content. While we present a snapshot of the V3Geo system
based on fast-tracked release to react to the COVID-19 pandemic, several key additions are in progress to further develop the
system for publishing 3D data in geoscience. Furthermore, we include guidelines for preparing model data in line with general
aims to ensure a baseline reliability and quality for the database. With the V3Geo initiative, we aim to increase accessibility
and adoption of virtual 3D geoscience, facilitating uptake and usage where large 3D datasets, associated reliance on high
performance computing hardware, and challenges with sharing application data have traditionally been a barrier for many.



**Appendix A: 3D model preparation and formatting for submission to V3Geo**

In this Appendix we provide guidelines for preparing 3D models ready for submission to V3Geo. Following these rules of
thumb will ensure efficient conversion and performance of contributed models in V3Geo, as well as ensuring high quality
database content for scientific and professional use. The aim is to provide practical considerations for model preparation rather
than a scientifically rigorous review of best practice.

**A.1 V3Geo visualization and model preparation**

The V3Geo web viewer uses tiled 3D models (processed with multiple levels of detail – LODs) to ensure efficient streaming
over the internet. Standard "single-file" models are not optimal, as they require all the model data to be accessed and stored in
memory prior to viewing. With today's huge datasets, even better spec PCs and good bandwidth will result in long download
times and inefficient performance without tiling. The authors have developed an algorithm to convert a standard 3D model
into the tiled format for V3Geo. To ensure this conversion runs successfully, we need to be aware of some common issues in
model processing, such as coordinate systems, model resolution and export format.


Concrete examples are provided based on the commonly employed photogrammetric software Agisoft Metashape (Agisoft,
2021). However, this does not reflect an endorsement by the authors; the described steps and considerations are similar in other
software, and are relevant to models processed using lidar, structured light scanning and more.

**A.2 Coordinate systems and model scale**

It is not a requirement that models are georeferenced, though it is an advantage for their later measurement and analysis,
especially if it will be combined with other 3D models, e.g. for making virtual field trips. It is recommended to use
georeferencing to position datasets acquired in the field (using on-board satellite positioning, external differential GPS or
ground control points depending on application requirements; see e.g. Eltner et al., 2016). Currently only projected (Cartesian)
coordinate systems are supported, not geographic (latitude/longitude) or geocentric. The coordinate system should be defined
accordingly during project setup, or the transformation applied on model export. It is recommended to use Universal Transverse
Mercator (UTM). Alternatively, a local coordinate system may be used if you have no georeferencing information, such as for
models of hand samples, fossils etc. For photogrammetric projects, model scale should be ensured through known ground
control or distance measurements identifiable within the imagery and 3D model.

It is recommended to use orthometric heights over ellipsoidal for the vertical datum (see vertical reference, below). Units
should be meters in X, Y and Z.





## A.3 Coordinate offsets

Because 3D graphics cards are often limited to single precision values, models with large coordinates (long numbers before
the decimal symbol) will cause jittery movement or "tearing" of the visual scene – especially noticeable when rotating.
Although this is handled by the V3Geo viewer, it is recommended to offset the *X* and *Y* values of the coordinates during model
processing and for export.

Offsets should be chosen to translate the model center to approximately (*0, 0*). It is practical to read coordinates in e.g.
Google Earth or other 3D software to get suitable offset values (such as by placing a pin in the center of the field area and
reading off the coordinates; Figure 6). The values can be rounded up or down to the nearest hundred or thousand. If using
Google Earth, set UTM readout coordinates in the Tools->Options dialog (Show Lat/Long). For example, using the
coordinates close to the model center marked in Figure 6 (*X, Y*): 369016.81, 5606654.21), the chosen offset is (*X, Y*):
369000, 5606600.


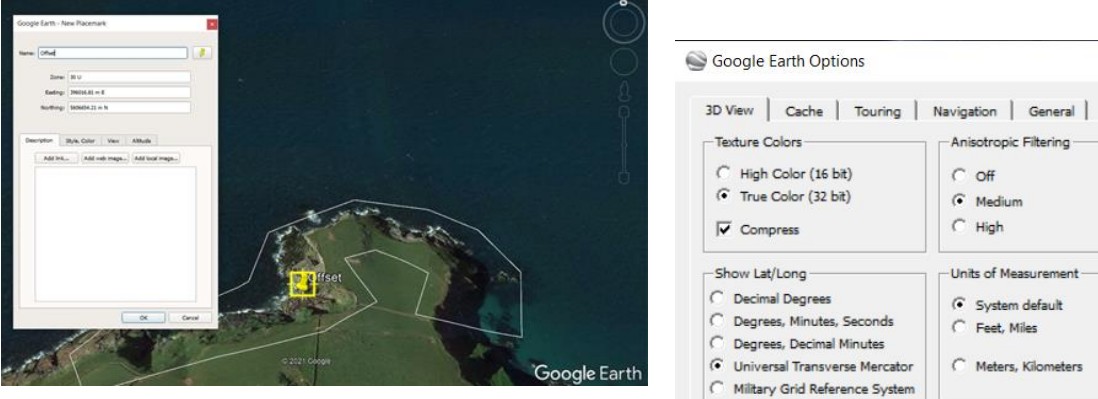

**Figure 6. Use a GIS or earth viewer such as © Google Earth to define coordinate offsets for model processing and export.**

- Record the offset – it will get reapplied during V3Geo conversion.
- It is important that the same offset is used for multi-section models.
- It is useful to use the same offset for 3D models that come from the same vicinity.

## A.4 Vertical reference

Satellite positioning using handheld devices (such as built-in to cameras or other acquisition devices) typically does not provide
high accuracy compared to differential positioning or ground control points. The vertical positioning is affected worst, resulting
in height values that may be off by meters or even tens of meters. It is advantageous to correct heights where possible. This
may be done in several ways to gain the *Z* offset between measured and real-world values:
- Using an existing map or terrain model (if common features can be identified);



- Reading off height coordinates in a GIS or virtual world software, e.g. Google Earth;
- Use mean sea level if field area is coastal.

The vertical offset can be applied during data processing or while exporting the model.

**A.5 Model size (triangles, texture)**

The best results come from smaller to medium-sized models. For most visualization and interpretation purposes, it is often not necessary to generate the densest point cloud and mesh possible. A mesh that is smoother, with well-distributed and fewer triangles is preferred over very dense and "rougher" meshes, though this is a guideline and is dependent on application. For practical purposes, aim for around 3 million triangles in the final mesh (Figure 7).


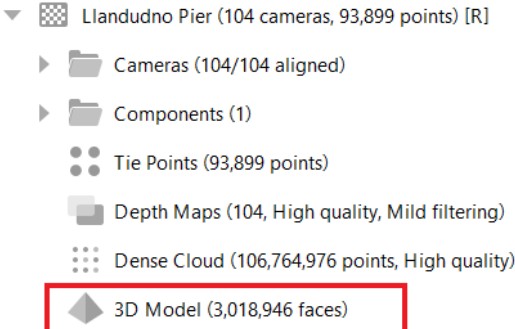

**Figure 7. Number of triangles in generated mesh (Agisoft Metashape).**

Considering the following points can help to achieve a suitable-sized mesh:

- Control the mesh size by splitting up the project area into multiple sections
- Be careful when generating the dense point cloud (and subsequent mesh) – selecting the highest resolution outputs will generate many millions of triangles, which is usually not necessary.
- Perform mesh decimation to have fine control on the number of model triangles. Software such as Meshlab is useful here.

Textured models have additional images that are generated to give the photorealistic texture. Again, care should be taken when
generating model texture that the texture resolution is not overly high. Similarly, default settings may result in low texture resolution if only a single texture is generated. The example in Figure 8 has only a single texture generated by default, which may result in a very low resolution texture unless the **texture count** is increased (must be reached empirically, but can be 10 or more). Ensure **texture size** is not greater than 4096 × 4096 for best compatibility across multiple platforms and devices.



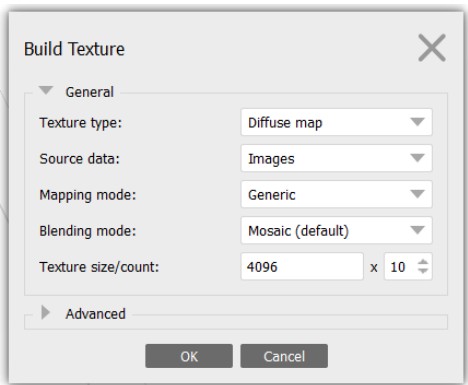

**Figure 8. Texture generation parameters in Agisoft Metashape.**

**A.6 Some simple steps to ensure well-formed, high quality models**

Processing point clouds of the natural environment, which may contain vegetation, noise, or other data artefacts, into meshes involves advanced algorithms and results may require some editing. Photogrammetric or lidar processing software often has in-built functionality for mesh cleaning, or specialized software (such as Meshlab) can be used to edit meshes, resulting in a

higher quality model prior to texturing. It is recommended to perform some basic mesh editing steps before texturing the model:

- Trim models according to the area of interest
- Check and trim boundary meshing artefacts
- Remove isolated triangle patches ("shells")

- Remove duplicate or very close vertices
- Clean self-intersecting triangles
- Minimize holes where possible

The presence of these artefacts, or a mesh that requires much editing, can be indicative of underlying problems with the data registration, causing a high level of point cloud noise or "double" surfaces, where the same area is modelled multiple times.

**A.7 Number of sections and overlap requirement**

If the field area is very large, or there is the requirement for a high level of detail to be modelled, it is recommended to split up the project into multiple sections (rather than be restricted to one lower resolution model). Web models for each section are combined during upload to V3Geo, allowing high resolution and large areas to be loaded in the V3Geo viewer. In this case, ensure some overlap between adjacent model sections is preserved (e.g. a strip with thickness of c. 20-50 triangles is usually

sufficient) to avoid distracting data gaps in the final model.



### A.8 Model export

Final model files should be exported as **Wavefront OBJ** (*.obj). This format is commonly available for read and write in 3D graphics software and is currently most "standard" to be handled by the tiled model converter. Export an OBJ file for each model section of the project considering the following (Figure 9):

- Projected coordinate system (use UTM where possible) and in meters
- Apply the coordinate offsets in *X* and *Y* (and *Z* where applicable)
- Avoid using vertex colors (not supported at this time)
- Export texture images as jpg
- Add an optional comment to the OBJ file (e.g. the coordinate offsets used)
- Avoid spaces in the chosen filename (use underscores "_" instead)

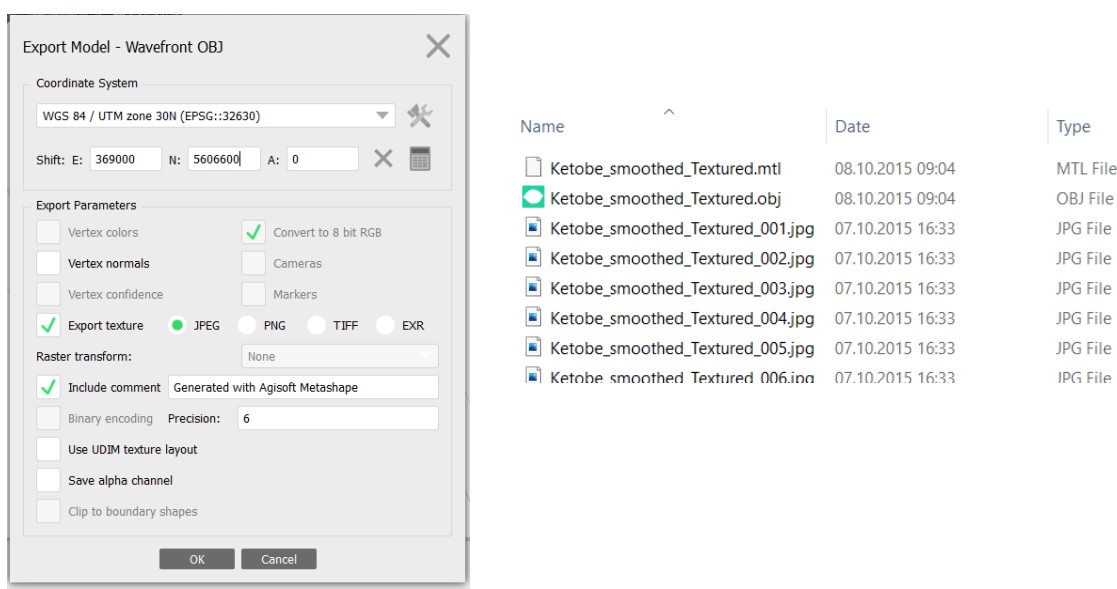

**Figure 9. Left) Export parameters recommended for V3Geo (Agisoft Metashape); Right) Disk file structure of output OBJ model, including associated .mtl and texture files.**

A textured OBJ model comprises several files – the .obj file itself, a material file (.mtl) and one or more image files (number equals **texture count**). When submitting OBJ models, all the related files need to be shared (use a zip archive to simplify transfer).


**Data availability**

Virtual outcrop models presented in this paper are available on V3Geo (https://v3geo.com).

**Author contribution**

JAH, NN and SJB conceived V3Geo, developed the specification and supervised the project. SJB and JAH wrote the main manuscript draft and established the reported usage scenarios. NN supervised V3Geo implementation and testing, and edited the manuscript. KR developed the V3Geo web viewer and early versions of the tiled model processing workflow. JV developed automated routines for tiled model processing. BT worked on authorisation and model upload capabilities. OSMT and DF implemented the V3Geo back end and website. MC, CL and GM prepared model submission workflows, formulated in the 580 Appendix, and worked on database content, including liaising with external contributors. JP implemented usage scenarios for V3Geo datasets in courses at University of Aberdeen. All authors read and gave input through multiple iterations of the manuscript draft.

**Competing Interests**

The authors declare that they have no conflict of interest.

**Acknowledgements**

The authors extend their thanks to the SAFARI consortium (https://safaridb.com) for continued support and encouragement to develop the idea of a public outcrop and 3D model repository for the geoscience community. OMT Tech AS is acknowledged for collaboration and implementation of the V3Geo back end infrastructure and main website. V3Geo is the culmination of several years of idea maturation and development. During this time, it has been discussed with and received feedback from 590 many members of the geoscience community, including the communities active around the Virtual Geoscience Conference series (https://vgc-series.org) and workshops on digital geoscience initiatives convened in Pau (November 2018) and Amsterdam (January 2020), which helped provide inspiration and community support for the V3Geo initiative. Finally, we acknowledge the groups from around the world who have been engaged with V3Geo through contributing models, providing feedback and support, as well as examples of their own usage scenarios.

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
