# Peer review of "V3Geo: A cloud-based repository for virtual 3D models in geoscience"

_Geoscience Communication, 2021_

## Author Comment (AC1)

**Reviewer 1**

It was a pleasure to read the article titled "V3Geo: A cloud-based repository for virtual 3D models in geoscience" written by Simon John Buckley, John Anthony Howell, Nicole Naumann, Conor Lewis, Magda Chmielewska, Kari Ringdal, Joris Vanbiervliet, Bowei Tong, Oliver Severin Mulelid-Tynes, Dylan Foster, Gail Maxwell, and Jessica Pugsley. In this article they present a tool/platform to make 3D geological data accessible for example for teaching, but also accompanying publications and just as shared data. The article is well written and without being aware of this upcoming publication, I actually used the V3Geo earlier this year to publish data for a publication. Hence, I first hand experienced the workflow and can confirm the suggested user experience in the article.

Dear Tobias,

Many thanks for taking the time to review our paper, and also for your positive response.

Nevertheless, I do have two minor comments (accepted subject to minor revisions), which I would like the authors to address:

Could the authors clarify what the difference (advantages/disadvantages) are between e-rock? Since e-rock, seems to be a very similar to V3Geo and thus in disagreement to the authors statement: "Although these studies highlight the practical benefits of web-based sharing of 3D models in geoscience, no current single repository has been presented for scientific and professional purposes. Solutions are limited in file (and therefore dataset) size, precluding many of the details needed for interpretation, do not allow supplementary interpretations or datasets, or are too broad in scope, covering all areas of society rather than being tailored to the geoscience community" (Line 72-75) Regarding the interpretation pf data sets – V3Geo seems to me from a user perspective not yet different to other solutions on the market. Is the data interpretation only possible via Lime or did I overlook some of the features. Please clarify.

V3Geo has several main advantages as we see them. A) the database can handle very large and high resolution datasets comprising multiple sub-model sections (https://v3geo.com/model/367 is a good recently published example, comprising 24 input sections each with around 3 million triangles). B) V3Geo has close to 300 contributions from around the world at the time of writing. C) Models are searchable based on the underlying data standard and database. D) Models get a basic quality control. E-Rock is a great initiative to create collections of virtual geology datasets, by embedding 3D models from Sketchfab within their own webpage with supplementary descriptions and diagrams, rather than a database. If desirable for the project, it would even be an option to embed the V3Geo models within relevant e-Rock pages to expand the collection of models available to the community as a portal with specialised focus. We have altered the sentence on e-Rock to reflect your comments. It is indeed important that the differencing is clear and to stress that both can coexist.

Regarding interpretations, V3Geo has display of interpretations built into the database, schema and 3D viewer. Upload of interpretation files and associating them with an author's model is currently work in progress at the time of writing. We plan to support interpretation overlays (most applicable to LIME in the first instance) and generic polylines, in line with the API reaching a stable release (mentioned in Section 2.3.5). Other data types will be considered in future updates.

The authors mentioned the V3Geo platform has been designed to handle different scales of model (microscopic, hand specimens, outcrop, etc. (See Line 11-12, 26-27) However, the text is only talking about outcrop-models. Will the focus stay on outcrop models? Will V3Geo take at the same time hand samples into account, making it more similar to e-rock? Similar, what about the microscopic

scale? Are these mentioned, because the platform could be used as a framework for similar data platform related to microscopic samples? Please explain, why the mention of microscopic when the text is all about outcrop scale?

You are correct to point out that the content of V3Geo at time of submission is skewed towards geological outcrops rather than hand samples etc. This reflects the authors' main field of research and general network. In the meantime, several hand samples have now been contributed (e.g. https://v3geo.com/model/343) and we have tested the system with both 3D fossil models and a 3D model generated from SEM data. There are also several landslide models in the database (e.g. https://v3geo.com/model/228), and we are in correspondence with other contributors outside of pure geology. We have made some minor adjustments of the manuscript to, we hope, provide a more balanced use of "3D models" rather than "outcrops".

---

## Author Comment (AC2)

**Reviewer 2**

This manuscript introduces an online repository for 3-D photographic models, aimed for research and science education, that additionally allows for user interaction and inspection. Thanks to advances in and wider availability of digital capturing techniques, 3-D photographic models are becoming a standard tool in field-related studies. Providing an online community repository is therefore a very useful and timely service. Once used and populated with more content, the platform has the potential to become an integral resource and tool to the wider Geoscience community. The platform's long-term viability, which is key in such a context, is discussed and clarified to the user. The manuscript is written very well and nicely illustrated. Whether the manuscript fits the journal's scope is up to the editor(s), but I highly suggest that Geoscience Communication also publishes manuscripts introducing community tools like this and not only research-based work. I therefore recommend this widely useful work for publication after some minor revision.

Dear Fabio,

Thank you for the time spent in reviewing our manuscript, and for your positive comments and suggestions.

Could it be clarified whether updates (of faulty data) or extensions (with additional related data) are possible at all, and whether those would be traceable by the users (e.g., via versioning)? Similarly, could a dataset be made obsolete, if another, better/newer one is published (to avoid duplication, continued use of outdated data, and save storage space)?

This is a very important point that was not directly addressed in the original manuscript. It is something we have had in mind during conception of V3Geo but have not yet addressed in the technical solution at time of publication. As it stands, V3Geo model entries do not have versioning. We have replaced some of our own early 3D model contributions to reflect improved processing workflows. However, we agree that such changes should be transparent to users. We have DOIs on the roadmap for V3Geo. This would enforce versioning, as model entries with DOIs will not be changeable following publication. We also see that some users will not want (or be ready for) DOIs. In that case a simple history of changes could be added. Creating a new model entry for an updated version is already an option. We have added a sentence to reflect your comment at the end of section 5.3 on publication.

Adding a short, simplified guide for unexperienced readers to the manuscript to explain how a 3-D photographic model is created (starting from standing in front of a sample/outcrop), is not key to the paper, but would, I think, be useful to promote their wider use and give unexperienced field geologists a feeling about whether they could use this tool too, or not.

The first draft of this manuscript actually included an additional appendix on drone-based photogrammetric 3D model acquisition (as one popular but not exclusive method) for creating 3D models. As the manuscript length increased, however, we decided to instead publish the acquisition guidelines as a separate preprint (Howell et al., 2021), which is referred to in the text. We believe it is out of the scope to go into much detail on the acquisition and processing methods themselves, simply because V3Geo is not restricted to one technique. We have added a sentence in the first paragraph of the Introduction with references for interested readers to consult about the two key methods of photogrammetry and laser scanning.

Line 9: Being a geodynamic modeller, reading the title and first sentence, my first impression was the manuscript being about an online repository of physical models of the Earth. I guess other readers

with yet another background might understand it to be about conceptual models. Given that you have a rather wide audience in this journal, maybe it would be worth clarifying the term ‚model' early on at the beginning of the abstract, by using a term like "3-D photographic models" or similar?

Thanks for raising this point from a different perspective. Indeed, use of "model" often requires some clarification whenever multidisciplinary themes are present. We have tried to limit the potential for misunderstanding by qualifying the term with "mesh" and "terrain" in the abstract and adding a bracketed definition at the start of the Introduction. In addition, the acquisition techniques photogrammetry and lidar are mentioned very early in both abstract and introduction. "Photographic" is unfortunately not precise enough, because you may have a 3D model that has been collected using e.g. laser scanning where there is no image texture, but still a perfectly valid 3D mesh model…

Lines 30-37: Is the use of 3-D photographic models also bringing down field-work costs in general (by reducing the work force and time in the field)? If so, would that be another very good argument to mention here?

Good point. We hesitate to say that use of virtual 3D models brings down the general fieldwork costs (for the team doing the acquisition it may actually increase it, depending on equipment and size of area chosen). However, facilitating sharing of this dataset, and reuse in research, teaching or the professional community, can be an important optimisation of resource use when sustainability is in focus in society. This was hinted at in the last sentence of the penultimate paragraph of the original manuscript's Introduction, but now expanded to explicitly state this:

"With a common sharing platform, this can avoid repeating acquisition efforts (and associated costs in time and resources) through routine sharing of high quality and reliable models, an important benefit when sustainability is in societal focus."

Lines 50-54: Out of interest: Is that a similar approach to the quadtree used in e.g., Google Earth?

Yes, Google Earth (and many others) use a similar approach for loading terrain/map/image tiles into its viewer. Quadtrees are traditionally associated with 2D or 2.5D data types, while octrees or other spatial segmentation structure are applied to 3D data.

Line 224: Consider clarifying to: „several gigabytes of model data".

Done